# Inflammatory Bowel Disease and Customized Nutritional Intervention Focusing on Gut Microbiome Balance

**DOI:** 10.3390/nu14194117

**Published:** 2022-10-03

**Authors:** Camilla Fiorindi, Edda Russo, Lucrezia Balocchini, Amedeo Amedei, Francesco Giudici

**Affiliations:** 1Department of Health Science, AOUC Careggi, 50134 Florence, Italy; 2Department of Experimental and Clinical Medicine, University of Florence, 50134 Florence, Italy

**Keywords:** diet, nutrition, microbiota, microbiome, inflammation, food, inflammatory bowel disease

## Abstract

Inflammatory bowel disease (IBD) represents a chronic relapsing–remitting condition affecting the gastrointestinal system. The specific triggering IBD elements remain unknown: genetic variability, environmental factors, and alterations in the host immune system seem to be involved. An unbalanced diet and subsequent gut dysbiosis are risk factors, too. This review focuses on the description of the impact of pro- and anti-inflammatory food components on IBD, the role of different selected regimes (such as Crohn’s Disease Exclusion Diet, Immunoglobulin Exclusion Diet, Specific Carbohydrate Diet, LOFFLEX Diet, Low FODMAPs Diet, Mediterranean Diet) in the IBD management, and their effects on the gut microbiota (GM) composition and balance. The purpose is to investigate the potential positive action on IBD inflammation, which is associated with the exclusion or addition of certain foods or nutrients, to more consciously customize the nutritional intervention, taking also into account GM fluctuations during both disease flare-up and remission.

## 1. Introduction

Inflammatory bowel disease (IBD) defines a group of heterogeneous chronic diseases of the gastrointestinal system characterized by a relapsing inflammatory process and alterations in the immune system [1]. Both the incidence and prevalence of IBD have rapidly increased over the years, to the point of assuming a significant weight globally [2]. IBD is thought to be caused by a combination of hereditary and environmental factors, leading to a dysequilibrium of pro- and anti-inflammatory mediators [3]. Environmental factors (EF) are deeply explored to investigate their role in both disease onset and risk for disease recurrences. EF can be variable and thus may be also a therapeutic opportunity. Among these, it now seems clear that diet plays a major role in IBD [4], influencing the developing risk and modulating the disease activity. Specific nutritional compounds are demonstrated to have positive effects on gut microbiota (GM) homeostasis, intestinal mucosa permeability and on the modulation of the inflammatory response [5]. Moreover, the GM influence on the pathogenesis and the progression of IBD has been extensively examined [6]. Mainly, a decreased abundance of *Firmicutes*, *Bifidobacterium* genus, and the *Faecalibacterium prausnitzii* species has been observed. In addition, a decrease in general diversity has been detected [7]. Therefore, the evaluation of a specific diet as a therapeutic option has increased interest, and several clinical studies have attempted to understand its impact, considering the potential effect on GM homeostasis, mirroring the need to find treatment options able to link patient needs with the decrease in disease-related complications. Considering the mechanisms of action underlying the effectiveness of different dietary patterns in IBD, two principal (and interconnected) patterns appear: (i) a potential direct regulation of immune response and (ii) a positive action on the GM (composition and function) and so on the degradation pathways of different food components. However, the weak evidence and some conflicting results did not lead to the definition of a single dietetic strategy. There are currently no dietary regimens to reduce symptoms or to induce/maintain remission that has been shown to be fully effective and universally appropriate [8]. At the same time, exclusion diets, if not appropriate, can increase the risk of IBD patients developing malnutrition, sarcopenia and micronutrient deficiencies; this can have a negative impact on quality of life and increase morbidity [3]. 

In our review, we first discuss the two main dietary approaches used in IBD: supplementation with anti-inflammatory components and exclusion/restriction of pro-inflammatory components, highlighting the pro- and anti-inflammatory activity of food elements and their complex reciprocal interactions with the GM. Then, we investigate the therapeutic indication of the main types of diets that have been studied as a potential nutritional therapy in inducing or maintaining the remission disease phase, with particular attention to the documented GM alterations induced by each specific dietary pattern.

Our purpose is to define the positive effects that are associated with the exclusion or addition of certain foods or nutritional components. Such evidence could allow us to better customize the nutritional intervention, considering the potential effect on GM, according to the professional’s critical thinking, as well as to the needs and clinical history of each IBD patient. To achieve this aim, the current review takes into consideration what has recently emerged from the literature on dietary approaches, GM, and IBD.

## 2. Goals of Nutritional Intervention

In general, the main goals of nutritional intervention in IBD patients can be framed into two macro categories: the active disease phase and the remission phase (Table 1). Therefore, nutrition targets will change depending on the patient’s current clinical situation, and their priority will be continually re-established based on the evolution of the disease.

To date, the study of dietary interventions in IBD patients with mild to moderate disease activity or in remission has primarily focused on two approaches that can be used independently or in combination: (1)Supplementation with anti-inflammatory components.(2)Exclusion/restriction of pro-inflammatory components.

## 3. The Gut Microbiome as a Mediator of Anti- and Pro-Inflammatory Food Components

Dietary nutrients and non-nutritive bioactive elements, such as minerals, fat, vitamins, trace elements, dietary fiber, alcohol, probiotics and prebiotics, as well as phytochemicals, affect immunomodulatory and inflammatory processes. In addition, the GM is able to biotransform food elements, producing metabolites, which alone or combined with food components can modulate the host immune system and metabolic responses. Indeed, many reports today point out the existence of reciprocal communication between the immune system and microbial flora, which has been reported as the “microbiome–inflammation or microbiota–immunity axis” [9,10]. Our immune system has evolved diverse strategies to tolerate the commensal bacteria maintaining homeostasis, so the mammalian commensal microbiome does not induce a pro-inflammatory response in a normal healthy contest. However, when this delicate equilibrium is impaired, due to perturbation by EF, damage of the interface between host and microbiome, or perturbations of the immune system, the consequences are (i) systemic propagation of commensal microorganism, (ii) susceptibility to pathogen infiltration, and (iii) aberrant immune response. 

In this context, IBD represents the archetypal disease in which the GM and the gut immune system lose their equilibrate interplay. A crucial contributor to the establishment of this balance is microbial competition for food nutritional components [11]. However, since microbiota architecture also depends on nutritional components, GM is frequently considered a mediator through which foods exert their pro- and anti-inflammatory actions (Figure 1). 

Moreover, animal models have shown that meals high in dietary heme [12], saturated fats [13], salt [14] and sugar [15], as well as deficient in fiber [11] can induce inflammation through microbiota pathways, such as the activation of T helper 17 (Th17) cells. Furthermore, other investigations have found that chemical substances inserted during food preparation, such as antimicrobial additives [16], dietary emulsifiers [17], and artificial sweeteners [18], enhance gut permeability and intestinal inflammation by increasing mucolytic bacteria and endotoxins. On the other hand, an increased amount of fibers [19] and tryptophan [20] often contribute to immunological responses associated with colonic health. 

Today, through functional experiments, we are learning more about the anti- and pro-inflammatory properties of single nutrients. Nevertheless, there is currently a lack of information on how dietary patterns and whole foods affect the GM and the host, and if these effects differ in the healthy vs. inflamed bowel. There have been several clinical studies focused on specific nutrients, in contrast to very few food-based interventions. Nonetheless, these studies do not consider nutritional interactions within the food matrix, which may explain the conflicting and restricted results reported [8]. Identifying the linking present in whole foods in relation to dietary patterns may lead to more efficient therapeutic indications for specific diets also in the IBD field.

## 4. Anti-Inflammatory Dietary Components and Microbiome in IBD

Foods with a higher anti-inflammatory potential can decrease the generation of pro-inflammatory factors, such as interleukin (IL-6, IL-1β, IL-8), tumor necrosis factor (TNF)-α, reactive oxygen species (ROS), nitric oxide (NO), and prostaglandins (PGs). The foods with recognized anti-inflammatory properties are green leafy vegetables such as kale; whole grains such as wheat berries, quinoa, and oatmeal; and fruits, tea, coffee, and wine. In particular, the most crucial anti-inflammatory elements found in these foods are ω-3 PUFA, carotenoids, flavonoids, terpenes, vitamins, and fiber. 

### 4.1. Anti-Inflammatory Plant-Based Foods

In detail, vegetables are rich in antioxidant vitamins A, C, and E and dietary fiber as well as minerals, phenolic compounds and trace elements. Oligo fructose and inulin, polymers of fructose found in artichokes, onions and asparagus, have been shown to promote commensal gut *Bifidobacteria* and *F. prausnitzii* in healthy individuals [21]. In subjects affected by irritable bowel syndrome (IBS), cruciferous vegetables can induce bloating; however, cauliflower, broccoli, Brussels sprouts, kale and cabbage produce indole-3-carbinol, which induces the activation of the aryl hydrocarbon receptor (AHR), which is a protein that links exogenous molecules such as natural plant flavonoids, polyphenolics and indoles. This protein also has been shown to be involved in regulating immunity. Experiments in mice reported that feeding with broccoli increased intestinal AHR activity, decreased *Erysipelotrichaceae* richness, and impaired colitis [22]. In addition, a component of cruciferous vegetables, called 3,3 Diindolylmethane (DIM), can restore the intestinal permeability of human intestinal cells [23]. Concerning the tubers, potatoes have large quantities of resistant starch with a pre-prebiotic activity. Specifically, the sweet potato family is rich in vitamins such as β-carotene and tocopherol, as well as dietary fibers, amino acids, and minerals [24]. Anti-inflammatory compounds such as anthocyanins, phenolic acids, polyphenols [25], and polysaccharides are also present [26]. A cross-sectional study on 103 adult patients (50 with active disease and 53 in remission, divided by their calprotectin level) who completed a self-administered food frequency questionnaire on their intake of several foods over 1 year showed protective effects of potatoes aggregated with legumes [27]. However, legumes might potentially exacerbate symptoms. In rats, feeding with red kidney beans, which are high in dietary fiber and resistant starch, resulted in changes in GM and cecal fermentation [28]. Furthermore, wheat bran is a cereal composed of non-starch polysaccharides (NSP) from β-glucans, cellulose, and arabinoxylans [29]. It is well absorbed by IBD patients and protects against colon cancer, which is a very common cancer among IBD patients [30]. Wheat includes gluten, which can worsen GI symptoms [31]. Wheat also contains amylase trypsin inhibitors that exert a detrimental action on the GM, promoting gut inflammation in animals [32]. Oats, on the other hand, are a carbohydrate-rich cereal that does not contain gluten but also present dietary fiber and beta-glucans. In particular, beta-glucans significantly decreased colitis clinical symptoms in mice by the downregulation of pro-inflammatory factor expressions such as IL-1β, TNF-α, and IL-6 in the gut [33]. On the other hand, rice elicited a considerably increased immunological response in CD patients [34]. Unrefined rice can boost helpful bacteria while lowering the amount of *Clostridium species* [35]. Additionally, quinoa and amaranth belong to the group of the so-called “superfoods” and have a nutritional composition that confers multiple benefits. Indeed, their seeds offer a high protein quality and dietary fiber content, as well as polyunsaturated fatty acids and anti-inflammatory polyphenols [36]. Quinoa consumption reduced intestinal dysbiosis by boosting species richness and variety as well as alleviating clinical symptoms in colitis mice [37]. In a study on human GM, these pseudocereals were subjected to an in vitro digestion and used as carbon sources in batch cultures with fecal human inoculate. Both quinoa and amaranth promoted the proliferation of some bacterial populations that increase acetate, propionate and butyrate levels, which are known as short-chain fatty acids (SCFAs) [38]. This research suggested that these pseudocereals can have the prebiotic potential and that their intake may improve dysbiosis or maintain the gastrointestinal health through a balanced intestinal microbiota. Moreover, mushrooms have been used as a kind of anti-inflammatory medicine since ancient times. A combined extract of *Basidiomycetes* mushrooms reduced inflammatory symptoms in IBD patients, resulting in interesting anti-inflammatory effects as demonstrated by declined levels of pathogenic cytokines in blood and calprotectin in feces [39]. Mushroom glucan can modulate cytokine profiles and phagocyte activity as well as improving protection against inflammation, infections, and sepsis in IBD [40]. Regarding soy protein, a soy-derived hydrolysate, high in di- and tripeptides, showed the inhibition of inflammation with a reduction in colon inflammatory cytokines’ production in colitis pig [41], while in colitis mice, treatment with a tripeptide resulting from enzymatic hydrolysis of soy reduced the production of pro-inflammatory factors and thus colitis manifestation [42]. A current report in IBD mice showed that the replacement of animal protein by soy reduced IBD severity [43]. Concerning fresh fruits, they are often high in fiber, vitamin C and other antioxidants such as phenolic acids, which are absorbed through the gut wall and may exert anti-inflammatory activity [44]. Apples, blueberries, citrus fruits, mangos, kiwis, and plums all contain a phenolic acid called chlorogenic acid, which is a significant and physiologically active dietary polyphenol that serves a variety of essential and therapeutic functions [45]. Grapefruit intake, on the other hand, frequently worsens symptoms, bringing severe distress to IBD patients [46]. Moreover, exotic fruits such as mango demonstrated lower levels of pro-inflammatory cytokines [47]. It is also high in gallotannin, which is a tannin that improves the microbial structure of the feces and reduce symptoms in colitis mice by inhibiting NF-B and MAPK signaling [48]. Inulin, one of the most important prebiotics, is found in bananas. Green bananas contain the type 2 granular resistant starch composed of amylose [49], which might be fermented into SCFA by GM [50]. Concerning dried fruits, nuts are high in unsaturated fatty acids, fiber, and protein as well as a variety of minerals (potassium, magnesium, and copper), vitamins (tocopherol, niacin, pyridoxine, or folic acid) and phytochemical components (stigmasterol, resveratrol campesterol and catechins). On the other hand, important antioxidants such as hydrolysable tannins, flavonoids, flavonols, anthocyanins, proanthocyanidins, stilbenes, flavanones, isoflavones, and phenolic acids are abundant in pecans, pistachios and walnuts [51]. Catechins, anthocyanins, and proanthocyanidins make up the majority of the polyphenols in cocoa. TLR 4/NF-B/signal transducer and activator of transcription are only a few of the signaling pathways that cocoa polyphenols activate. In fact, they also affect the GM, causing bacteria to proliferate and activate the anti-inflammatory pathways of the host [52]. Both monounsaturated and saturated fatty acids are found in cocoa butter [53]. In addition, copper, magnesium, iron, and potassium are abundant in chocolate and cocoa [54]. Chocolate intake, on the other hand, was linked to an augmented incidence of Ulcerative Colitis (UC) and Crohn’s Disease (CD) [55]. However, in a randomized controlled trial, cocoa intake lowered NF-κB in peripheral blood mononuclear cells (PBMCs) in healthy individuals, implying a reduction in the production of pro-inflammatory cytokines [56]. In colitis mouse models, treatment with cocoa-derived polyphenols resulted in a considerable decrease in TNF-α and IL-1 levels in the inflamed colon [57]. Analogously, feeding these murine models with cocoa diet reduced blood TNF-α, colon cell infiltration and inducible nitric oxide synthase activity, indicating anti-inflammatory potential [58]. Furthermore, the administration of cocoa polyphenols in mice reduced nitric oxide (NO) production, neutrophil infiltration, and the expression of STAT-1, COX-2, and STAT-3 as well as drop of IL-6, IL-1β, and TNF-α from peritoneal macrophages [59]. In vitro cocoa polyphenols treatment of intestinal cells induces an increase in prostaglandin E2 production via cyclooxygenase (COX)-1 action, which is important in preserving the gut mucosal barrier [60]. Artificial enteral nutrition enriched in polyphenols might prevent or increase IBD inflammatory status [61]. Caffeine and chlorogenic acid are found in coffee [45]. Caffeine can downregulate the expression of chitinase3-like 1 (YKL-40), producing anti-inflammatory effects in the gut of colitis-induced animals [62]. Moreover, polyphenols from green tea operate as antioxidants and have anti-inflammatory properties by inhibiting the production of NF-kB, TNF-, IL-1, and other inflammatory factor as well as building a healthy GM, regulating claudin and occluding [63].

Moderate wine drinking is linked to antioxidant benefits, which are attributable to a wide range of phenolic compounds [64]. Wine phenolics are free radical scavengers as well as modulators of genes related to inflammation and cellular redox signaling [65]. In addition, phenolics can decrease isobutyrate levels [66] and may lower dangerous bacteria while stimulating good GM components such as *Bifidobacteria* and *Lactobacteria* in an in vitro study [67]. Moreover, extra virgin olive oil is especially emphasized due to its phenolic compounds, which can decrease radical oxidative species (ROS) and prevent inflammatory diseases. In addition, it is able to alleviate the symptoms of chronic inflammation by blocking arachidonic acid and NF-kB signaling pathways [68]. These phenolic elements are missing in oils obtained from seeds or fruits [69]. Indeed, diets based on olive oil decrease inflammation more than diets based on seed oils, indicating that oleic acid is more anti-inflammatory than linoleic acid [70]. Extra virgin olive oil decreased the expression of TNFα gene in the colonic mucosa of rats [71]. Furthermore, the unsaponifiable component of olive oil enhances apoptosis and decreases the stimulation of T cells isolated from IBD patients, influencing receptor integrin 7 expression on blood T cells [72]. Herbal products are widely used by IBD patients. Polyphenols are abundant in dried herbs such as thyme, oregano, and basil. Ginger and cumin have also anti-inflammatory properties [73]. Notably, curcumin, a phenol derived from turmeric, inhibits nuclear factor-κB, signal transducer, p38 mitogen-activated protein kinase, and Th1 cytokines in human intestinal microvascular endothelial cells [74].

### 4.2. Anti-Inflammatory Animal-Source Foods

Regarding animal products, fish oil demonstrated anti-inflammatory properties in IBD, as it contains molecules involved in the regulation of immunological and inflammatory responses as eicosapentaenoic acid (EPA), long-chain n-3 polyunsaturated fatty acids (PUFAs), and docosahexaenoic acid (DHA) [75]. However, there are not enough data to suggest that n-3 PUFAs could be used in clinical practice. [76]. Honey consumption has a number of health benefits for CD patients [46]. Acacia honey and citrus fruit contain acacetin, which is a bioactive component flavonoid that is able to reduce macrophage infiltration, perhaps enhancing its therapeutic impact in IBD. Moreover, fermented dairy products, such as cheese and yogurt, induced anti-inflammatory effects in colitis mice [77]. Milk kefir, a fermented dairy product, induced a decrease in C-reactive protein [78]. A study in colitis mice revealed that feeding with the emmentaler (a particular cheese) fermented by *Propionibacterium freudenreichii* reduced IgA secretion in the small bowel, preventing the induction of TNFα, IFN-γ, and IL-17 [79]. A peptide from buffalo milk regulated the NF-kB pathway and reduced intestinal permeability in the DNBS-induced colitis. Butter is derived from milk fat containing butyrate, which is a fatty acid that is also generated by GM. Butyrate is able to hamper inflammation by boosting NF-kB and colonocytes in human [80].

## 5. Pro-Inflammatory Dietary Components and Microbiome in IBD

Meat is surely a good source of protein, vitamin B12, and iron, but it also includes saturated fat. Nevertheless, meat and especially processed meat (the latter prepared by smoking, curing, salting, or preservative addition) [81] has also a high concentration of organic sulfur and sulfate additions. This may increase the amount of sulfate available for bacterially produced hydrogen sulfide. Protein fermentation by-products, notably H2S, phenols, and ammonia, have negative impacts on the colonic milieu and epithelial health [82]. A high total animal protein consumption was linked to a higher IBD risk, while meat consumption was linked to a high risk of IBD recurrence. A carnivorous eating pattern was linked to an increased risk of developing [83] and relapsing of UC. Regarding eggs, currently, there is inadequate information about their relationship with the risk of CD and UC [84]. In CD patients, the immunological response to eggs was greatly elevated [34]. In DSS-mice models, the consumption of egg whites improved the disease activity index and weight loss as well as decreased the release of pro-inflammatory cytokines TNF-α and IL-6 [85]. Dairy products were not linked to IBD risk or illness recurrence in IBD [27]. They are high in calcium, proteins, and riboflavin. Calcium is required in IBD patients to avoid metabolic bone damage [86]. Lactose, the sugar of the milk, enters the colon and causes bloating and/or diarrhea when it is not digested owing to lactase insufficiency. Animal models revealed that diets high in saturated milk fat favored *Bacteroidetes* while suppressing *Firmicutes* [87]. Furthermore, this diet stimulated the growth of *Bilophila wadsworthia*, which is thought to be pro-inflammatory, due to its sulfite-reducing and immune activating and capabilities. Moreover, a high milk fat diet expedited the development of colitis in mice by boosting the taurine conjugation of bile acids, and it also increased luminal sulfur availability and the subsequent proliferation of *Bilophila wadsworthia*.

Concerning the vegetables, tomato is a common cause of dietary sensitivity in IBS patients [88]. In CD patients, the immunological response to tomato was significantly enhanced [34]. Moreover, non-starch polysaccharides (NSP) from cellulose, arabinoxylans, and glucans make up about 46% of wheat bran [29]. Although it may benefit IBD patients, wheat may potentially cause GI symptoms, as observed in CD remission [89]. In addition, wheat includes gluten (gliadin), which can increase GI symptoms including stomach pain, bloating, and diarrhea even independently from celiac disease [31]. 

Among vegetal fats, sunflower oil is mostly made up of linoleic acid (polyunsaturated fat) and oleic acid (a monounsaturated fat), and it is high in vitamin E [90]. It has been shown to increase the disease activity index and pro-inflammatory cytokine levels in mice [91]. Moreover, margarine consumption is associated with an increased UC risk [92], which may be because of the high quantity of linoleic acid, an n-6 PUFA, and trans fatty acids [93]. Regarding PUFA, a recent article demonstrated experimental evidence that an excess of dietary polyunsaturated fatty acids in a Western diet triggers metabolic inflammation in the gut, which may deteriorate the course of CD (PMID: 35031299).

High sugar and soft drinks intake is linked with the risk of UC [15]. In detail, cola drinks and saccharose were, respectively, linked to an increased risk of IBD and CD development [55,94]. Indeed, high saccharose intake augmented tissue inflammation in animal studies [95]. Furthermore, excessive saccharose consumption was linked to endoplasmic reticulum stress, which was similarly linked to CD risk [96] but not UC [97].

In terms of salt intake, in animal colitis models, the contact of intestinal cells with a high NaCl intake promoted the generation of some inflammatory cytokines, such as IL-17 and IL-23, in normal intestinal *lamina propria* [98]. Dietary salt stimulates the intestinal Th17 response and macrophages, and it decreases in the meantime the suppressive actions of Tregs in mice models [99], promoting an inflammatory environment [100]. Furthermore, a high salt diet aggravates colitis and has a negative influence on gut microflora by lowering *Lactobacillus* numbers and butyrate synthesis.

Finally, many processed foods include emulsifiers and thickeners [101]. The small intestine is the place where many emulsifiers are demolished by digestion, but condensing agents (carboxymethylcellulose and carrageenan) may be harmful throughout the gastrointestinal tract [101], leading to changes in GM composition, bacterial inclusion into mucus, and thus the development of chronic inflammation in human [102]. Maltodextrin is a common food ingredient used as a thickening or sweetener. A higher maltodextrin intake was associated with a significant increase in CD incidence [103]. Notably, higher amounts of *E. coli* and adherent-invasive *E. coli* strains were found in ileal CD patients, implying an *E. coli* role in disease pathophysiology [104]. Maltodextrin enhances adherent-invasive *E. coli* adherence and biofilm development on the intestinal CD epithelium [105]. These data indicate that consuming maltodextrin may promote *E. coli* colonization and contribute to CD vulnerability [106]. 

The relevant findings of anti-inflammatory and pro-inflammatory dietary components on GM in IBD are reported in Table 2.

## 6. Therapeutic Indications for Specific Diets and Their Effect on Gut Microbiome

As previously reported, IBD occurrence and development depend on the interplay between several factors such as genetics, EF exposure, including food, having a negative or positive action. In this context, diet has received considerable attention as a readily modifiable EF and, as a result, as a potential preventative or therapy for IBD. Nutritional treatment includes a variety of diets often based on the exclusion of specific food groups. Which diet to choose is a controversial topic. The criteria for choosing nutritional intervention are often not reported in the available studies. Generally, the nutritional approach may concern type of disease (CD, UC, chronic pouchitis), disease activity, clinical history, nutritional status, and individual food intolerances.

However, no current guidelines encourage a particular diet that should be maintained throughout remission or active illness.

Among all nutritional therapy, in this paragraph, we examined the dietary interventions recommended the most in IBD patients, taking into account, when possible, their effect on gut microbiome composition.

### 6.1. Severe Disease’s Activity and Nutritional Recommendation

In case of a severe disease, the nutritional intervention must be elaborated according to the presence of complications, preventing malnutrition until the clinical conditions improve. A diet with adapted texture or distal EN (enteral nutrition) can be recommended in CD patients with intestinal strictures or stenosis in combination with obstructive symptoms [8]. Many studies focus on how EN can mitigate negative effects and diminish pathobionts. Changes in microbiota composition have been seen to be rapid, emerging even within one week from receiving EN treatment [107]. EN seems to lower microbial diversity, SCFA concentrations, and *Faecalibacterium prausnitzii*, all of which are generally thought to be favorable in IBD [108]. Moreover, responders to EN had lower bacterial richness than those who did not respond [109]. In a recent prospective study of children initiating partial enteral nutrition (PEN), exclusive enteral nutritional (EEN), or anti-TNF therapy for CD, clinical outcomes were compared using the Pediatric Crohn’s Disease Activity Index (PCDAI), QOL (IMPACT score), and mucosal healing as estimated by fecal calprotectin (FCP). PCDAI, IMPACT, FCP, and diet (prompted 24 h recall) were measured at baseline and after 8 weeks of therapy. EEN induced a decrease in Shannon diversity, but this returned to pretreatment levels two months after the EEN stop, as did decreases in *Bifidobacterium*, *Ruminococcus*, and *Faecalibacterium* [110]. However, the likelihood of therapeutic response might be determined by how the baseline microorganisms adapt to such a severe diet change [111]. Patients who showed reductions in *Bacteroides* and *Prevotella* species with EN treatment improved clinically, and these GM modifications lasted for several months after EN treatment was completed [112]. 

In addition, PN is suggested in several conditions, for example when both oral nutrition or EN are not applicable, when the intestine is obstructed and a feeding tube has not been placed beyond the blockage, or when additional problems such as an anastomotic leak or a high output intestinal fistula arise [8]. Nutritional support is always indicated in patients with malnutrition. The PN effects on the GM is very poorly understood. A recent study examined features of GM and bile acid (BA) metabolism in inducing the capacity to wean from PN [113]. The authors observed variations in the microbial populations of short bowel syndrome (SBS) patients with ileostomy compared with jejunostomy, jejunocolonic compared to ileocolonic anastomoses, and PN reliance against those who were weaned off PN. Stool and serum BA composition changed in SBS patients, indicating an alteration in enterohepatic BA cycling. Secondary BAs such as deoxycholic acid and lithocholic acid were found in the stools of individuals who had been weaned from PN. Changes in gastrointestinal microflora and BA metabolites may develop a positive luminal microenvironment in certain patients, allowing them to wean off PN.

### 6.2. Nutritional Intervention in Moderate to Mild Disease’s Activity 

#### 6.2.1. Crohn’s Disease Exclusion Diet (CDED)

Another type of diet that has stood out in recent years is the “Crohn’s Disease Exclusion Diet” (CDED), which is a partial EN associated with an exclusion diet. It has been studied mainly in pediatric/adolescent patients; there are also positive results in adults, but the available studies include a small sample of subjects. 

CDED is based on the exclusion of foods that can increase the intestinal permeability and change the microbiota in a pro-inflammatory direction. The foods potentially responsible for these effects are those with reduced fiber intake, excessive fats intake (especially saturated), simple sugars and maltodextrins, foods rich in taurine, emulsifiers, preservatives, carrageenan, gluten and sulfites. 

It has an efficacy in inducing the phase of remission equal to the exclusive enteral nutrition, with the advantage of being more tolerated by patients, since in addition to the polymeric formula, it also includes the consumption of natural foods. It is important that the patient follows the instructions provided, as it has been shown that a partial enteral nutrition associated with a free diet does not lead to the same results in terms of disease remission. 

The CDED is divided into three different phases: the first two (lasting both six weeks) and the third phase of maintenance, which starts from the thirteenth week. During all three phases, part of the caloric and nutrient requirements is ensured by EN, which is a liquid, nutritionally complete and adequate polymer formula, free of fiber and lactose. In the first phase, the energy and nutrient intake from EN is 50%, decreasing to 25% in the second and third phases. In association with the EN, there is a diet with natural foods that is standardized and divided into mandatory, permitted and prohibited foods. 

There are more studies in pediatric patients than in adults. Although CDED can be considered an effective therapeutic strategy during the active phase in adult patients who do not respond optimally to drug therapy, strong evidence is lacking [114]. In children with mild to moderate CD, CDED plus partial enteral nutrition and exclusive enteral nutrition were both successful in determining remission after six weeks of treatment [107]. CDED in combination with partial enteral nutrition appears to result in a high incidence of remission in early mild-to-moderate luminal CD [115]. 

Changes in microbial structure and function were correlated to clinical response in CDED. The CDED diet, when combined with EN, was more tolerated and more effective in producing remission than EN alone [107]. Both diets had a comparable impact on bacterial activity over the first 6 weeks: a decrease in bacterial richness of *Actinobacteria* and *Proteobacteria* and increase in commensal *Clostridia* was observed. Nonetheless, between 6 and 12 weeks, these alterations were found inverted in CDED +EN, while in CDED + PN, the microbial composition remained similar [107]. These modifications in the microbial flora during the therapy may suggest a better outcome of CDED + PN compared to PN alone. Moreover, the trial of Sigall-Boneh et al. corroborated the efficacy of the CDED diet [115]. 

#### 6.2.2. Immunoglobulin Exclusion Diet (IGED)

The antibodies produced in response to the intake of food antigens can also be detected in healthy subjects who do not develop symptoms related to ingestion; numerous studies have shown that the antibodies’ production is a physiological response to foods regularly ingested, following previous exposure to that food [116]. 

Studies have shown that IBD patients had higher levels of responses to specific food antigens than healthy subjects. The implicated subclasses of anti-food antigens antibodies were mostly IgG. In particular, IgG1 is an initial responder, while IgG4 is produced after a chronic antigen exposure. For this reason, the elimination of foods with a high antigen content has been studied and tested as a nutritional approach for IBD patients [117]. According to these criteria, the most commonly excluded foods were eggs, cow’s milk and dairy products, beef, pork, and wheat [118]. The aim of these diets was to improve the symptoms, reducing the level of inflammatory markers. 

A pilot study attempted to assess the role of foods with high IgG antibody levels and additives on CD symptoms and inflammation, inferring that foods with elevated IgG antibody levels and food additives can aggravate clinical signs and may enhance inflammation in CD patients. According to the findings of the study, an appropriate diet with limitations on particular items may be advantageous in medical therapies [119]. 

A recent prospective study included 97 UC patients, documenting that an IgG-guided exclusion diet improved UC symptoms and quality of life. However, interactions between IgG-based food intolerance and UC need further studies [120]. 

Furthermore, a double-blinded randomized controlled trial has evaluated the ability of an IgG4-guided diet in ameliorating the quality of life in patients with CD, concluding that this diet can improve both quality of life and symptoms [118]. 

However, the effect of IGED on GM composition and function has never been studied until now. As there is a mutual interplay between the microbiome and immune response, it would be useful to obtain data to verify the implications on the microbiome balance.

#### 6.2.3. Specific Carbohydrate Diet 

The Specific Carbohydrate Diet (SCD) was created as a therapy for the celiac disease and, given its favorable outcomes on UC, it was recommended as a further approach for IBD management. 

The premise behind SCD is that complex carbohydrates (disaccharides and most polysaccharides) requiring enzymatic digestion reach the colon partially digested, potentially promoting the fermentation and overgrowth of bacteria and yeasts and altering the microbial flora, contributing to a condition of intestinal inflammation. Contrarily, monosaccharides, not requiring enzymatic digestion and consequently limiting intestinal fermentation, can be included in this diet. The SCD purpose is to decrease intestinal inflammation by restoring the balance of the bacterial flora [121]. 

Unprocessed meats, most fresh vegetables and fruits, all fats and oils, aged cheeses, and lactose-free yogurt are all permitted foods. Milk, grains, soft cheeses, and non-honey sweeteners are all foods to avoid. When returning to a free diet, foods that were previously excluded are reintroduced one type at a time. 

A recent observational study has suggested that SCD has the potential to be a powerful tool in the treatment of some IBD patients, particularly those with colonic and ileocolonic CD. However, we need further evidence suggesting that SCD can alter the luminal environment, particularly the gut microflora, and consequently, SCD could be considered an effective treatment for some patients with IBD [121].

Notably, SCD is among the few diets with some, albeit limited, evidence of a beneficial effect on mucosal healing as well as clinical manifestations. However, the SCD’s dietary restrictions call into question patients’ levels [122,123]. 

A potentially specific microbial effect for the SCD was observed in a 20-year-old female with UC [124]. After one year of SCD, the fecal microflora differed significantly from that of three healthy controls, suggesting a strong loss of bacterial diversity. Another change in bacterial composition was an increase in *Enterobacteriaceae*, which included *Escherichia* and *Enterobacter* species. After two SCD weeks, the prevalence of the most dominant fecal bacterial species decreased by two to threefold. Although species richness remained low, overall, the species diversity increased to levels comparable to controls. 

Moreover, a study conducted in 2014 on the GM of eight CD participants reported that patients having the SCD (contrary to those receiving a diet low in plant fiber) presented an increased GM diversity. The microbial families overrepresented in the intestinal tract comprised more than 20 species of non-pathogenic bacteria of the *Clostridia* family [125]. In addition, Hoffman showed that carbohydrate consumption can rise *Candida* and *Methanobrevibacter* archaea, which produce simple carbohydrates, using starch. These carbohydrates are substrates for *Prevotella* and *Ruminococcus*, which produce metabolites that *Methanobrevibacter* needs to generate methane and carbon dioxide [126].

#### 6.2.4. Low Fat Fiber Limited Exclusion Diet 

The Low-Fat Fiber Limited Exclusion (LOFFLEX) diet was developed after the observation that patients with Crohn’s Disease often experienced a worsening of symptoms when returning to a normal diet (after following an exclusive enteral diet to induce remission), especially with the reintroduction of foods with higher fat and fiber content [127]. This dietary protocol provides, once a state of remission is reached, a food reintroduction plan by eliminating foods less tolerated by patients [128].

Therefore, the LOFFLEX Diet provides the limitation of fats (in particular long-chain triglycerides) to 50 g/day and the reduction in fiber to 10 g/day due to excessive fermentation by the intestinal flora that may be induced by fiber. The protocol also requires the exclusion of some foods, such as yeast and coffee, which can be associated with unwanted symptoms. 

The LOFFLEX diet should be followed for about 2–4 weeks, and during this time, a food diary should be used to accurately identify the source of any symptoms that may develop. 

An inadequate number of studies are available to correlate the LOFFLEX Diet with a real efficacy as a nutrition strategy for IBD patients. Currently, in the literature, no information is available on the effect of this diet in modulating the microbiome

#### 6.2.5. Low Fodmaps Diet

About 35% of IBD patients experiences gastrointestinal symptoms such as abdominal pain, bloating, flatulence, and diarrhea, even during the remission phase of the disease. These functional symptoms often meet the diagnostic criteria for irritable bowel syndrome (IBS), indicating that patients have a higher prevalence than the general population [129,130].

The presence of such intestinal disorders may negatively affect the quality of life; various dietary patterns have been proposed as a potential treatment approach to limit gastrointestinal symptoms. The most effective strategy recognized for the management of intestinal symptoms in IBS patients is a diet with reduced content of FODMAPS, or fermentable oligosaccharides, disaccharides, monosaccharides, and polyols. The main FODMAPs found in foods are fructose, lactose, fructans, galactans, and polyols. Some of these molecules have osmotic properties and increase small intestinal water volume, while others are not completely digested in the small intestine and undergo bacterial fermentation in the colon, resulting in gas production [131]. 

These processes could be responsible for the onset of symptoms or its worsening in patients with gastrointestinal hypersensitivity. The dietary strategy includes a first phase of elimination of foods rich in FODMAPs, a second phase of gradual reintroduction according to the patient’s tolerability and finally a maintenance phase. In the food reintroduction phase, it is useful to use a food diary to monitor the onset of any symptoms following the intake of the foods initially excluded.

A randomized controlled trial was conducted to assess the effects of a low FODMAP diet on prolonged gut symptoms, and the results showed that a low FODMAP diet for four weeks is safe and efficacious for handling persistent gastrointestinal symptoms in patients with quiescent IBD [132]. 

A different clinical study showed that a short-term, low FODMAPs diet was linked with improved fecal inflammatory markers and quality of life even in patients with mainly quiescent disease [133].

According to other studies, a low FODMAPs diet may be useful in patients with IBD remission but with gastrointestinal symptoms to reduce these symptoms and improve their quality of life [134,135]. 

The elements eliminated in this diet include prebiotics, which can specifically increase the development of *Bifidobacterium* and *Faecalibacterium prausnitzii* and are substrates for the production of SCFA [136]. A low-FODMAP dietary intervention is able to induce a decrease in both the relative and absolute quantity of butyrate-producing bacteria. Fructooligosaccharides may boost *Bifidobacterium* abundance and improve the immunological function of dendritic gut mucosa cells. [137]. Moreover, they are able to reduce the pro-inflammatory cytokines, e.g., IL-6, and increase that of the anti-inflammatory cytokine IL-10 [138]. A low-FODMAP diet has a considerable influence on GM; in fact, following a 4-week intervention, the overall numbers of bacteria may drop up to six times [139]. Furthermore, the low FODMAP diet reduced the total amount of gut microorganisms., especially of *Clostridium cluster* IV, including *Faecalibacterium prausnitzii*, *Bifidobacterium*, and *Lactobacillus* [140]. In addition, *Bifidobacterium* is reduced [139]; in fact, a decreased abundance of *Bifidobacterium adolescentis* and *Bifidobacterium longum* was detected in recent studies [132]. *Faecalibacterium prausnitzii*, *Ruminococcus* spp., and *Bifidobacterium longum*, which are members of the starch degradation process, can affect diet interventions in IBD [141]. 

Finally, in patients with quiescent CD or UC and intestinal symptoms, following a low FODMAPs diet, a significantly lower abundance of *Bifidobacterium adolescentis*, *Bifidobacterium longum*, and *Faecalibacterium prausnitzii* was observed when compared to those on a control diet [132]. 

#### 6.2.6. Mediterranean Diet

The Mediterranean Diet (MD) involves the intake of phytonutrients contained in plant foods such as fruits, vegetables, and antioxidant compounds: for example, omega-3 polyunsaturated fats contained in fish and nuts, unsaturated fats such as olive oil, but also high-fiber whole grains and a low consumption of red meat. Adherence to MD is related to a reduction in inflammatory markers and therefore may be promising as a possible strategy to be applied with IBD patients. There are evidence that this dietary pattern reduces the incidence of both CD and UC [142], but the effect on maintaining remission has not yet been well investigated. However, unlikely other dietary approaches that involve the exclusion of various foods, MD is less likely to cause nutritional deficits in an IBD patient. 

Disease activity, quality of life, and the patient’s surgical history in IBD all impact adherence to MD. Increased adherence to MD as a result of nutritional education may aid in increasing quality of life and regulating disease activity [143].

In particular, the assumption of legumes, fruits and certain varieties of vegetables containing soluble fibers has a prebiotic effect and, consequently, can favor the growth of microbial species with production of some SFCA, such as propionic acid and butyric acid and favoring the decreased secretion of inflammatory cytokines and improvement of intestinal permeability [144].

As shown by some studies, the adherence to this diet can positively influence the intestinal microbiota of healthy subjects, and some positive results on disease activity and nutritional status have also been observed in IBD patients [145,146]. A food model based on the Mediterranean Diet can also be useful as treatment for overweight/obesity in IBD patients, reducing their BMI and improving some biochemical parameters [147,148].

In a recent research on healthy Italians, eating habits and GM were examined. It was observed that high levels of adherence to MD have a positive influence on the GM and related metabolome [149].

The intake of fruits, vegetables, and legumes by subjects who showed an acceptable adherence to the MD was linked to a growth in fecal SCFA. This result was most likely helped by bacteria from the *Firmicutes* and Bacteroidetes, which are capable of degrading indigestible carbohydrates. In addition to GM variations, CD patients following the MD for 6 weeks exhibited a change in the expression of more than 3000 genes [146].

Because the MD contains a high fiber content, it may be inappropriate for individuals during disease flares, but it is strongly recommended following remission with major modifications [150,151]. In reality, the usage of pulses, which contain soluble fibers, has a prebiotic impact, supporting the increase in microbiota species producing propionic and butyric acid, which reduces the production of inflammatory cytokines.

#### 6.2.7. Vegetarian Diet

Vegetarian diets (VD) are described by the high consumption of fruits, vegetables, legumes, nuts, and grains, minimizing processed foods and animal foods. A semi-vegetarian diet allows including eggs and milk, small weekly portions of fish, and meat once every two weeks. Thanks to its high fiber intake, this diet is assumed to be protective against mucosal inflammation due to the immunomodulating role of SCFAs [152]. Consequently, some studies have addressed the association between vegetarian diet and IBD development. The beneficial effects on health are believed to be due to plant polyphenols in the diet, which have anti-inflammatory, cardioprotective, chemoprotective, antiestrogenic, and neuroprotective roles.

Currently, there is not enough evidence to recommend it universally. It is important to evaluate the needs/habits of the individual patient in order to prevent nutritional deficits, which may be often associated to VD if it is not adequately planned by a dietitian [153,154].

The GM in vegetarians has been linked to enhanced bacterial diversity, with notably reduced counts of pathobionts *Enterobacteriaceae* and increased members of *Prevotella* [155]. A recent cross-sectional study on MD and VD found that short-term dietary intervention does not induce significant changes in the GM, implying that a diet should last longer than 3 months to scratch the microbial resilience [156]. Eventually, the GM composition of meat-eating IBD patients was considerably different from those adopting a VD [154]. Many microbial alterations were detected in meat-eating IBD patients compared to those who follow VD or GFD, including reduced species richness with a dose–response impact in meat-eating CD patients. Surprisingly, the reverse trend was found in UC patients, with increased species richness in individuals who consumed meat more often. In conclusion, Figure 2 clarifies the potential therapeutic indications for specific diets based on the available above-mentioned evidence.

## 7. Conclusions

Several evidence underlie the interplay between diet, illness, and microbiome in IBD, suggesting that the disease origin and course is strictly connected with gut dysbiosis. In this narrative review, the relationship between selected diets and clinical course of IBD was analyzed, focusing on their effects on microbiome balance. However, as IBD is an inflammatory disease, it is mandatory to describe the pro and anti-inflammatory action of plant-based and animal-source foods on disease process and, as there is a mutual interplay between the microbiome and the immune response, it is crucial to assess their effects on bacterial flora architecture. Therefore, we report that healthy diet can be effective in maintaining remission and preventing flare-ups in IBD and also the nutritional therapy that leads to a significant impact on IBD management.

Nevertheless, due to a lack of scientific evidence, no current guideline is able to recommend the specific diet for uncomplicated IBD with the consequence that patients tend to independently modify their diet in order to control symptoms, but self-imposed food limitations are usually damaging to patients’ nutritional status, and the supervision of a nutritionist seems required. In fact, an individualized nutritional treatment is more appropriate for IBD patients affected by such a chronic and heterogeneous condition.

CDED and SCD could be an option during the transitional period from active phase to remission in order to control inflammation and gastrointestinal symptoms. On the other hand, LOFFLEX and FODMAPs diets are potential dietary treatments of functional symptoms that persist even during remission. Industrialized foods seem detrimental in every phase of IDB course. All these suggestions lead to promoting the achievement of a balanced and varied diet, which is identifiable in the Mediterranean Diet model, which is known for its anti-inflammatory effects.

A wider and well-designed research focused on the relationship between diet and microbiota during nutritional interventions seems required.

## Figures and Tables

**Figure 1 nutrients-14-04117-f001:**
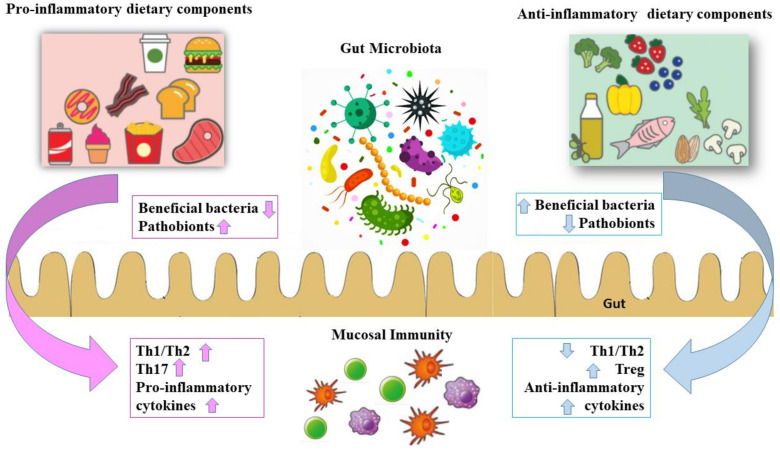
Diet plays a critical role in intestinal homeostasis, which is defined by the gut microbiota, intestinal mucosal barrier, and mucosal immune system. Diet directly modulates the mucosal barrier and immunity, whereas diet–microbiota interaction also regulates intestinal homeostasis. The up arrow indicates an increase while the down arrow indicates a decrease.

**Figure 2 nutrients-14-04117-f002:**
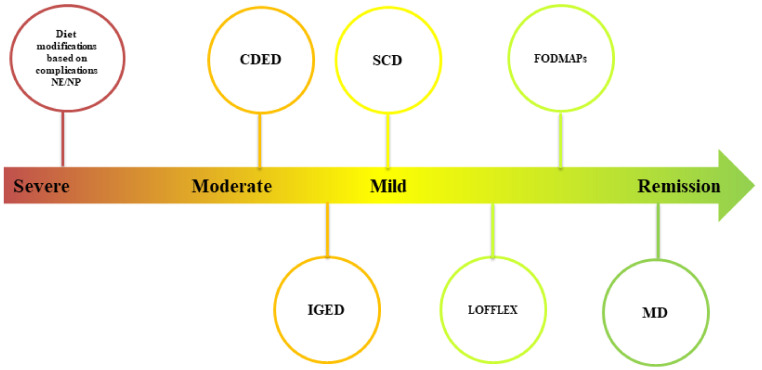
Potential therapeutic indications for specific diets based on disease severity. NP: parenteral nutrition; NE: enteral nutrition; CDED: Crohn’s Disease Exclusion Diet, IGED: Immunoglobulin Exclusion Diet; SCD: Specific Carbohydrate Diet; LOFFLEX: Low-Fat Fiber Limited Exclusion; MD: Mediterranean Diet.

**Table 1 nutrients-14-04117-t001:** Aims of nutritional intervention according to disease’s activity.

Active Phase	Remission Phase
▪ Prevent/treat malnutrition ▪ Prevent muscle loss and sarcopenia ▪ Avoid water and electrolyte imbalances and avoid dehydration ▪ Reduce mechanical irritation and promote bowel rest ▪ Replenish nutrient reserves in case of malabsorption ▪ Prevent nutritional deficiencies ▪ Prevent/treat anemia ▪ Modify the diet according to the presence of complications ▪ Improve nutritional status before surgery if indicated	▪ Maintain a good nutritional status by avoiding nutritional deficiencies ▪ Modify the diet according to the patient’s medical history (e.g., considering previous abdominal surgery) ▪ Modify the diet according to the presence of intestinal symptoms ▪ Educate the patient to follow a complete and balanced diet to maintain remission ▪ Improve the food-related quality of life ▪ Prevent osteoporosis in case of long-term use of corticosteroids by ensuring adequate intake of vitamin D and calcium ▪ Prevent specific nutritional deficiencies considering past surgical interventions

**Table 2 nutrients-14-04117-t002:** Relevant findings of anti-inflammatory and pro-inflammatory dietary components on GM in IBD.

Anti-Inflammatory Dietary Components and GM	Pro-Inflammatory Dietary Components and GM
Oligo fructose and inulin of artichokes, onions, and asparagus promote commensal gut *Bifidobacteria* and *F. prausnitzii* in healthy individuals [21].	Diets high in saturated milk fat favored *Bacteroidetes* while suppressing *Firmicutes* [87] and stimulated the growth of *Bilophila wadsworthia*. A high milk fat diet expedited the development of colitis in mice by boosting taurine conjugation of bile acids as well as increasing luminal sulfur availability and the subsequent proliferation of *Bilophila wadsworthia*
Feeding mice with broccoli increased intestinal AHR activity, decreased *Erysipelotrichaceae* richness, and impaired colitis [22].	A high salt diet aggravates colitis and has a negative influence on gut microflora by lowering *Lactobacillus* numbers and butyrate synthesis
Unrefined rice can boost helpful bacteria while lowering the amount of *Clostridium species* [35]	Emulsifiers in the small intestine are demolished by digestion, leading to changes in GM composition, bacterial inclusion into mucus, and thus the development of chronic inflammation in humans [102]
Quinoa consumption reduced intestinal dysbiosis by boosting species richness and variety as well as alleviating clinical symptoms in colitis mice [37].	Maltodextrin enhances adherent-invasive *E. coli* adherence and biofilm development on the intestinal CD epithelium [105]. Higher amounts of *E. coli* and adherent-invasive *E. coli* strains were found in ileal CD patients, implying an *E. coli* role in disease pathophysiology [104]
Quinoa and amaranth have prebiotic potential and improve dysbiosis or maintain the gastrointestinal health through a balanced intestinal microbiota. They promoted the proliferation of some bacterial populations that increase acetate, propionate, and butyrate levels [38].	
Green bananas contain the type 2 granular resistant starch composed of amylose [49] that might be fermented into SCFA by GM [50].	
Polyphenols from green tea have anti-inflammatory properties by inhibiting the production of NF-kB, TNF-, IL-1, and other inflammatory factors as well as building a healthy GM, regulating claudin and occludin [63].

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
