# Peer review of "Inflammatory Bowel Disease and Customized Nutritional Intervention Focusing on Gut Microbiome Balance"

_nutrients, 2022, doi:10.3390/nu14194117_

Round 1

Reviewer 1 Report

Fiorindi and Russo et al. narrative review aims a. to clarify the effects on the gut microbiota of exclusion or addition of certain foods or nutritional components in IBD patients b. to hypothesize potential indications for some nutritional therapies in different IBD settings.

Nutritional therapies in IBD are a hot topic. 

Anti-inflammatory dietary components and microbiome in IBD and Pro-inflammatory dietary components and microbiome in IBD chapters are clear and comprehensive. However, I would recommend reporting recent controversial results on PUFA (Schwärzler J et al Gastroenterology 2022). I suggest authors summarize relevant findings in a table for easier recall.

Therapeutic Indications for Specific Diets and their effect on the gut microbiome. 

- I suggest the authors reduce their statements' strength of recommendation…even with “potential” hypotheses are presented as “indications”. I believe suggestions could be enough based on the actual evidence. 

- I would recommend authors to include the statement “no current guidelines encourage a particular diet that should be maintained throughout remission or active illness” in this paragraph rather than in the conclusions.

- Authors should clarify the inclusion/exclusion criteria of dietary therapies examined or add all dietary treatments already studied in IBD patients (Table 3 Fitzpatrick JA Nat Rev Gastroenterol Hepatol. 2022). 

- Generally, the nutritional approach may concern … type of disease (CD, UC, chronic pouchitis)

- Authors should expand the abstract statement “assessing dysbiosis before recommending a dietary pattern in IBD should become a common therapeutic to obtain a tailored approach”

- Authors should correct paragraph numbering and some incorrect references.

Author Response

1) Anti-inflammatory dietary components and microbiome in IBD and Pro-inflammatory dietary components and microbiome in IBD chapters are clear and comprehensive. However, I would recommend reporting recent controversial results on PUFA (Schwärzler J et al Gastroenterology 2022). I suggest authors summarize relevant findings in a table for easier recall.

We thank the reviewer for appreciate our work, we inserted the recommended article from lines 301 to 303. We also inserted a table with relevant findings of Anti-inflammatory and Pro-inflammatory dietary components and microbiome in IBD (Table 2)

2)  I suggest the authors reduce their statements' strength of recommendation…even with “potential” hypotheses are presented as “indications”. I believe suggestions could be enough based on the actual evidence.

Thank you for the suggestion, we tried to reduced the statements strength of recommendation

3) I would recommend authors to include the statement “no current guidelines encourage a particular diet that should be maintained throughout remission or active illness” in this paragraph rather than in the conclusions.

We have included the suggested statement from line 344 to 345

4) Authors should clarify the inclusion/exclusion criteria of dietary therapies examined or add all dietary treatments already studied in IBD patients (Table 3 Fitzpatrick JA Nat Rev Gastroenterol Hepatol. 2022).

We clarified why we examined only the selected dietary therapies

5) Generally, the nutritional approach may concern … type of disease (CD, UC, chronic pouchitis)

We corrected it in the text

6) Authors should expand the abstract statement “assessing dysbiosis before recommending a dietary pattern in IBD should become a common therapeutic to obtain a tailored approach”

We rewrote the abstract taking into account both the revisions of the two revirwers, so the mentioned phrase has been erased

7)  Authors should correct paragraph numbering and some incorrect references.

We thank the reviewer, we corrected the mentioned issues

Reviewer 2 Report

nutrients-1914897_major revision

The manuscript entitled with “Inflammatory bowel disease and customized nutritional intervention: therapeutic indications focusing on gut microbiome balance. ” by Camilla Fiorindi, Edda Russo and other authors, was recently submitted to “Nutrients” as a review article for possible consideration. Overall, this manuscript is not focused but may be still suitable for the scope of the journal, well, several issues should be carefully addressed before the further step.

Major Issue 01: The title was interesting but confusing. It seemed to be attractive but not focused. The title however was mixed with several items together. I guess the authors wanted to describe healthy diets more than the nutritional intervention but general food nutrients rather than therapeutics on IBD, however, the descriptions of customized nutritional intervention or therapeutic treatment and gut microbiome were a little bit superficial. The scientific drawings and schematics were inadequate. The contexts between Line 502 to Line 552, there were just lists of the cited literatures.

Major Issue 02: In this manuscript, the authors declared that their “purpose is to define the positive effects that are associated with the exclusion or addition of certain foods or nutritional components”, therefore this review is some sort of biased studies. Both sides of the so-called dietary nutrients or the anti-/pro-inflammatory componence should be equally treated.

Major Issue 03: The “Abstract” was not good abstract for the whole manuscript. This section should be concise and clear. Figure 1 and Figure 2 were too simple to readers. More or less, the tables and the figures should support the main conclusions of the manuscript. The “Conclusion” was not that confident. If the authors wanted to prove that what they believed in faith, more organized data and informative figures should be provided.

Major issue 04: In fact, when I first read the title, I truly desired to read more and know more on this topic. After reading through this manuscript, I have to say, it was hard to understand the relation between nutritional intervention and IBD, and difficult to follow the clue of gut microbiome balance. It would be great if the authors could give an in-depth analysis or comparison. I suggest the authors to add more comments and good discussion in this manuscript. As you may notice, from line 136 to line 234, there is a superlong paragraph, talking about the known vegetable components. I just wonder, could the superlong single paragraph be divided into several small paragraphs?

By the way, in the manuscript, how can first two authors share the co-first authorship but not the corresponding authorship? Did they truly contribute equally in this case? And, only one reference was from the year 2022, regarding if it’s necessary to update the reference citation.

I wish to check up the revised manuscript and hope the quality of the manuscript after revision can be improved. My current recommendation for this manuscript is: Major Revision.

Author Response

 Major Issue 01: The title was interesting but confusing. It seemed to be attractive but not focused. The title however was mixed with several items together. I guess the authors wanted to describe healthy diets more than the nutritional intervention but general food nutrients rather than therapeutics on IBD, however, the descriptions of customized nutritional intervention or therapeutic treatment and gut microbiome were a little bit superficial. The scientific drawings and schematics were inadequate. The contexts between Line 502 to Line 552, there were just lists of the cited literatures.

-We thank the reviewer for its appropriate suggestions. The title was modified accordingly. We regret that the reviewer found the study design inadequate. We tried to improve the content of the review by following the suggestions of both reviewers. The intent of the review is to investigate customized nutritional intervention in IBD, focusing on the maintaining of gut microbiome balance. However, our approach can seem superficial as the literature on all the effects of IBD recommendend diets on microbiota alteration is not very thorough, at the moment, but it is a field still in exploration.  So, we thought about writing groundbreaking work that might be relevant to the journal. In addition, as there is a mutual interplay between microbiome and immune response, even more important in IBD, we though to describe also the effect of food on inflammation (pro and anti-inflammatory effects),  to have a broader view of food-inflammation-microbiome interactions.

 Major Issue 02: In this manuscript, the authors declared that their “purpose is to define the positive effects that are associated with the exclusion or addition of certain foods or nutritional components”, therefore this review is some sort of biased studies. Both sides of the so-called dietary nutrients or the anti-/pro-inflammatory componence should be equally treated.

 Dear reviewer, our manuscript is a narrative review focused on the different diets adopted in IBD patients, describing the goals of the nutritional intervention. We believe our work could be of interest for clinicians since not many similar studies are present in literature so far. It is true that we were not able to completely analyze both dietary nutrients and anti-/pro-inflammatory componence but it is quite difficult to report this kind of data in an adequate number of pages to be easily read and majorly avoiding to report misleading data as those obtained by experiences performed in general population and not in IBD. We hope you can agree with our purpose as we appreciate your comment.

Major Issue 03: The “Abstract” was not good abstract for the whole manuscript. This section should be concise and clear. Figure 1 and Figure 2 were too simple to readers. More or less, the tables and the figures should support the main conclusions of the manuscript. The “Conclusion” was not that confident. If the authors wanted to prove that what they believed in faith, more organized data and informative figures should be provided.

We modified the abstract accordingly to the rigt suggestion of the reviewer. However, Figures 1 and 2 are voluntarily extremely simple as they are a sort of key message to be easily recorded by different clinical figures (M.Ds, Dietitians, Nurses), while tables report data which were difficult to express and underline in the main text. Conclusion section was modified accordingly.

 Major issue 04: In fact, when I first read the title, I truly desired to read more and know more on this topic. After reading through this manuscript, I have to say, it was hard to understand the relation between nutritional intervention and IBD, and difficult to follow the clue of gut microbiome balance. It would be great if the authors could give an in-depth analysis or comparison. I suggest the authors to add more comments and good discussion in this manuscript. As you may notice, from line 136 to line 234, there is a superlong paragraph, talking about the known vegetable components. I just wonder, could the superlong single paragraph be divided into several small paragraphs?

We thank the reviewer for the indications, we rewrote the conclusion and we splitted the superlong paragrah into two pararaphs, as suggested. We also previously explained our intent to decribe the relation between nutritional intervention and IBD, and the clue of gut microbiome balance in our Major issue 01 answer.

 By the way, in the manuscript, how can first two authors share the co-first authorship but not the corresponding authorship? Did they truly contribute equally in this case? And, only one reference was from the year 2022, regarding if it’s necessary to update the reference citation.

-Yes, both authors contributed equally to the manuscript preparation, but only one (E.R.) was the one who performed the manuscript submission becoming the corresponding author. About references they reflects the actual more relevant literature available on this topic in IBD setting.

Round 2

Reviewer 2 Report

My concerned issues have been properly addressed and necessary improvements have been made accordingly. The quality of this manuscript by authors has been promoted and the current version looks much better. 

I have no further major questions and I am satisfied with the responses and answers from the authors.

Herein I may suggest the present revised manuscript to be accepted.